# Towards Encountered-type Haptic Interaction for Immersive Bilateral Telemanipulation

### Yaesol Kim
yaesol.kim@iit.it
Istituto Italiano di Tecnologia (IIT)
Genova, Italy

### Sara Anastasi
s.anastasi@inail.it
Istituto Nazionale per l'Assicurazione
contro gli Infortuni sul Lavoro (INAIL)
Rome, Italy

### Nikhil Deshpande
nikhil.deshpande@iit.it
Istituto Italiano di Tecnologia (IIT)
Genova, Italy

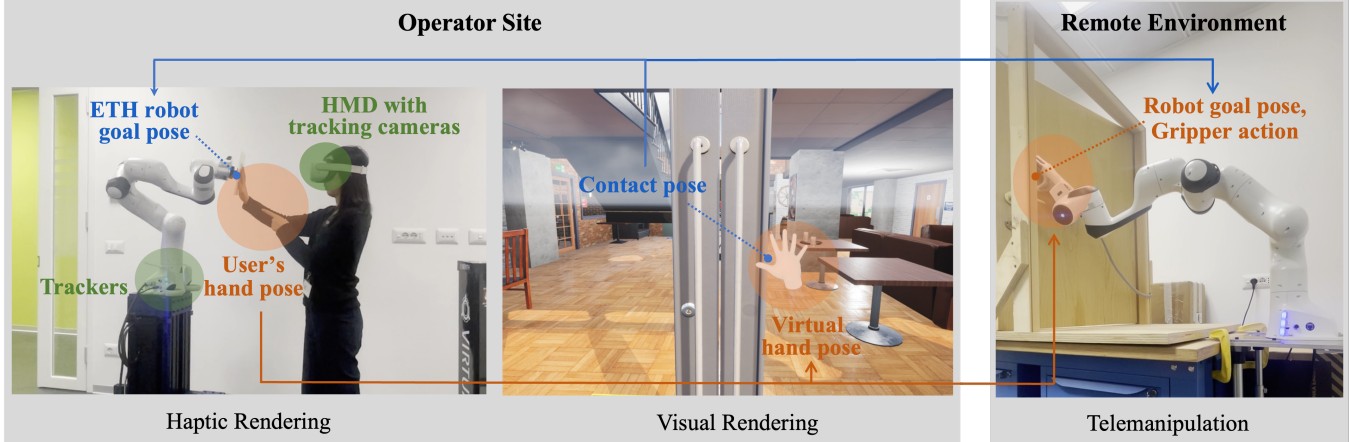

Figure 1: The encountered-type haptic interaction for telemanipulation. The human operator interacts with remote environment using one's bare hand with haptic feedback (left) and visual feedback (center). The robot on remote environment is telemanipulated based on the contact configuration and operator's hand pose.

## ABSTRACT

Encountered-type haptics (ETH) is an emerging research field allowing unencumbered physical haptic interaction in virtual reality (VR). In this paper, we introduce ETH as an interaction medium for immersive remote telemanipulation, to enable intuitive bare-hand interaction with visuo-haptic feedback. The proposed system enables an operator to telemanipulate a remote robot immersively using the visual rendering of the VR environment, while interacting with a real robot at the operator site. The ETH feedback rendering and the telemanipulation at the remote site are both implemented using the 7 degrees-of-freedom (DoF) Franka Emika robot under Cartesian-impedance control. The cartesian goal poses of each robot are decided according to the operator's interaction intention, estimated through the hand gesture and gaze direction tracking, and the rendered remote VR environment. The paper successfully demonstrates an ETH interface that provides intuitive and immersive interaction in immersive bilateral telemanipulation scenarios.

## CCS CONCEPTS

• **Computer systems organization** → **External interfaces for robotics**; • **Hardware** → **Haptic devices**; • **Human-centered computing** → **Virtual reality**.

## KEYWORDS

Immersive Telemanipulation, Encountered-type Haptics, Virtual Reality

**ACM Reference Format:**

Yaesol Kim, Sara Anastasi, and Nikhil Deshpande. 2023. Towards Encountered-type Haptic Interaction for Immersive Bilateral Telemanipulation. In *Proceedings of Workshop at ACM/IEEE HRI 2023 (VAM-HRI'23)*. ACM, New York, NY, USA, 4 pages.

## 1 INTRODUCTION

In an era marked by the Covid-19 pandemic, ongoing conflicts, and climate disasters, there is a growing interest in remote robot control as a means of performing tasks in hazardous environments, without exposing human operators to the associated dangers. To improve the user experience and task performance in remote teleoperation, immersive frameworks have been proposed by adopting the *virtual, augmented, and mixed-reality* (VAM) interface paradigm [10, 12, 13]. VAM-Teleoperation has been shown to enhance the presence and situational awareness of users through intuitive rendering of the visual and haptic cues of the remote scene in an immersive virtual

*VAM-HRI'23, March 2023, Stockholm, Sweden*

reality (VR) environment [15, 18]. Depending on the task condition, haptic rendering can enhance the teleoperation performance, significantly more than using just the visual cue rendering [8].

For the haptic interaction, due to the nature of the VAM interfaces in remote telemanipulation, the operator is usually required to wear a visual display, e.g., a head-mounted display (HMD), and is restricted to using commercial haptic rendering devices. The desktop-type haptic devices provide sophisticated force feedback, but the intuitive and seamless teleoperation is limited because the operator keeps holding the device and because of the device's small workspace. Hand-held or wearable haptic devices guarantee the operator's natural interaction, but they are limited in the rendering of the force cues from the remote environment. On the other hand, an encountered-type haptic (ETH) display or robotic graphics enables an intuitive VAM interaction by utilizing a robotic manipulator to deliver haptic feedback to the user [9, 19]. ETH interface allows the user to experience "free-to-touch" and "move-and-collide" haptic sensations without requiring the user to wear a haptic device nor mount it on the user's body [1, 3, 14, 16].

In this paper, we introduce ETH as a novel interaction interface in immersive VAM-telemanipulation. Integrating ETH with telemanipulation and supporting intuitive bare-hand interaction can enhance the operator's presence, immersion, and telemanipulation task performance. To the best of our knowledge, this is a first-of-its-kind approach in using ETH as an interface for bilateral teleoperation. This paper shows a first implementation of the ETH-based immersive bilateral remote telemanipulation interface and demonstrates the interaction scenario.

## 2 SYSTEM OVERVIEW

As illustrated in Fig. 1, our immersive ETH-Telemanipulation system follows a typical haptic teleoperation system, consisting of an operator site and a remote robot site [11]. For the operator site haptic interface, the following ETH rendering setup, based on the H-wall system from [4], is implemented:

(1) A virtual environment is created consisting of objects that the user can interact with and a user's virtual proxy hand is visually rendered in the scene, mapped in real-time with the real user hand.
(2) The 7-DoF Franka Emika manipulator robot arm operates as a haptic manipulator device (i.e., "ETH robot") to provide physical feedback to the user.
(3) By tracking a user's hand, our system estimates the user's intention of how and where they would like to interact with the virtual environment.
(4) The corresponding contact configuration of the virtual object relative to the user is estimated.
(5) A real fabricated object, modeled on and serving as a proxy for the virtual object, is attached to the end-effector of the ETH robot. The ETH robot alters the proxy object pose, following the human hands, thereby providing ETH feedback. The ETH robot relies on contact forces exerted by the user.

For the bilateral remote telemanipulation setup, the operator VR environment itself serves as a proxy for the real-world remote scene. The remote site consists of the real-world models of the objects shown in the operator site VR environment, e.g., the door with

handle. The estimated user intention, contact predicted configuration, and the exerted forces from the operator site are mapped to the remote manipulator robot arm (7-DoF Franka Emika arm, "remote robot"), thus making the remote manipulator follow the operator's actions. The subsequent motions of the remote robot and the forces exerted by it on the remote objects are rendered, through visuo-haptic rendering, back to the operator.

Both the robots are controlled using Robot Operating System (ROS). For the operator site setup, the Oculus Quest2 HMD was used to visualize the VR environment and its 4 monochrome built-in cameras were used to track the user's hand. Two Oculus Rift controllers were used as trackers for tracking the ETH robot base pose, and registering it with respect to the tracked user. For the proxy of the virtual or remote objects, we designed and printed 3D models (e.g., door handle) and attached it to the ETH robot end-effector. The VR setup was constructed using Unreal Engine 4 and the communication between the operator and remote sites was implemented using Rosbridge [2]. For the remote robot, to imitate the user's hand gestures, a robotic hand called Hannes [7] was attached to its end-effector.

## 3 METHOD

### 3.1 Coordinate Frame Rectification

To merge the three different worlds including the real world of the operator site, the real world of the remote robot site, and the virtual world, the coordinate system should be unified into a singular coordinate system [10, 17]. Since each world has multiple agents having their own coordinate frames, we calculate the relative transformation between agents' coordinate frames from each world and then place the virtual ETH robot and virtual remote robot in VR space.

To place the virtual ETH robot in VR scene, we calculate the relative pose $^C\xi_{R_{eth}}$ of the ETH robot coordinate frame $\{R_{eth}\}$ w.r.t. the HMD camera coordinate frame $\{C\}$. $^C\xi_{R_{eth}}$ is obtained through the two Oculus Rift controllers being placed on the left and right sides of the ETH robot base (refer Fig. 1). Since our current system implemented without vision system at the remote environment, the relative pose $^{R_{remote}}\xi_O$ of the physical object $O$ w.r.t. the remote robot coordinate frame $\{R_{remote}\}$ is predefined. Based on $^{R_{remote}}\xi_O$, the virtual remote robot is placed in VR scene.

### 3.2 Interaction Intention Estimation

To reduce the delay of ETH rendering and telemanipulation, the operator's intention about how and where to interact should be estimated. The type of interaction that the operator intends to have, such as navigation and object manipulation in VR, is estimated through the operator's bare-hand gesture. Since the frequent role of the dominant hand is manipulating the object and the role of the non-dominant hand is simpler than the role of the dominant hand [6], we assign a different interaction gesture for each hand. The dominant hand gesture is assigned to object-manipulating interaction such as grasp, release, and push, and the non-dominant hand gesture is assigned to VR navigation, such as turning left/right and moving forward. To this end, we recorded customized hand poses for each VR navigation action using hand-tracking data. Then, at runtime, the distance between the current operator hand pose

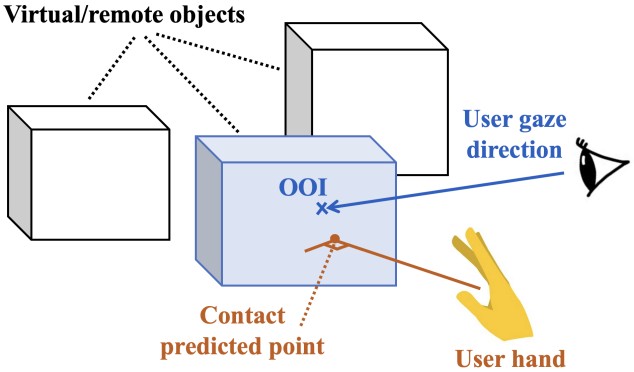

**Figure 2: Contact configuration prediction**

and the recorded hand poses is compared. When this distance is lower than a preset threshold, the corresponding customized hand pose is selected and the navigation action assigned to it is selected as the operator's intention. As shown in Fig. 1, the operator can use one's non-dominant hand to move forward in VR and push the door using one's dominant hand.

To predict the oeprator's intention about where to interact, we use the operator's gaze direction and the center of the palm of the operator's dominant hand [5]. As shown in Fig. 2, the system finds the object-of-interest (OOI) by casting the ray along the operator's gaze direction and selecting the closest hit object. The contact position is predicted by projecting the 3-DoF center of the palm of the operator's hand to the surface of OOI. The contact surface orientation is predicted as the surface orientation of OOI at the contact predicted position.

### 3.3 Bilateral ETH Telemanipulation

For the bilateral haptic telemanipulation, compliance control was used both for the ETH robot and the remote robot, to impose a dynamic environmental condition. Our system uses a Cartesian-impedance controller to control the position and impedance of the robots simultaneously. The robot end-effector dynamics are modeled as a mass-spring-damper system:

$$\mathbf{F_{ext}} = M\ddot{e} + D\dot{e} + Ke \tag{1}$$

where $M$, $D$, and $K$ are the mass, damping, and stiffness matrices of the system respectively, and $\mathbf{F_{ext}}$ is the external wrench from the user or environment. $e$ is $\dot{x} - \dot{x}_d$, $x$ and $x_d$ are the current and desired Cartesian poses of the robot end-effector.

The desired Cartesian pose of the ETH robot is decided based on the contact force exerted by the operator. The torque sensors embedded in each joint of the ETH robot are used to estimate the external force $\mathbf{f_{eth}}$ in a standard way:

$$\tau = \mathbf{J^T f_{eth}}, \tag{2}$$

where $\mathbf{J}$ is the manipulator Jacobian. When $\mathbf{f_{eth}}$ is lower than a pre-defined contact force threshold, the desired Cartesian pose is assigned to the contact predicted pose w.r.t. $\{R_{eth}\}$ as described in Sec. 3.2. Otherwise, desired Cartesian pose is not updated and ETH robot provide haptic feedback to the operator with $\mathbf{F_{ext}}$. Practically,

we assign the stiffness to the virtual/remote object and update the stiffness matrix at runtime depending on OOI.

The desired Cartesian pose of the remote robot follows the operator's hand pose w.r.t. $\{R_{remote}\}$. The contact predicted pose can be used for reflecting the user's intention to robot control, in a way that keeps the distance from the contact predicted point until the operator make physical contact with the ETH robot. To adjust the impedance of the remote robot according to the OOI, we update the stiffness matrix at runtime to the assigned stiffness to the remote object.

## 4 DEMONSTRATION AND DISCUSSION

This section shows the interaction scenario for the immersive ETH bilateral telemanipulation setup where the operator interacts with the ETH robot, while being immersed in a VR environment, and telemanipulates the remote robot to interact with the real-worl remote environment. The scenario includes a VR environment with a virtual door and the remote environment consists of a real door. The operator opens the virtual door by interacting with the proxy door surface object mounted on the ETH robot. The ETH robot provides the tactile (door surface) and haptic (robot force reflection) feedback. This door opening gesture is mapped to the remote robot, which pushes open the real door in the remote environment, operated through egocentric telemanipulation.

As shown in Fig. 3, the physical ETH feedback is provided to the user when the operator attempts to push the door visualized on the HMD. At the same time, the remote robot follows the user's action to open the door at the remote site. The egocentric telemanipulation using our system allows operator to immerse oneself in a virtualized remote environment and telemanipulate the robot intuitively.

## 5 CONCLUSION

In this paper, we proposed an immersive bilateral telemanipulation interface with encountered-type haptic feedback. By integrating the ETH interface with the telemanipulation system, our interface provides intuitive egocentric telemanipulation and increases immersion in VR and remote environments. We show the use case of our interface and demonstrates the interaction scenarios.

Our current method for bilateral haptic telemanipulation uses the predefined stiffness value assigned according to the virtual/remote object. Future work will handle runtime mapping algorithms between an external force from a remote robot and a operator site haptic force. Further investigation will also include the analysis of the effect on user immersion, task performance, and situational awareness compared to telemanipulation using commercialized controllers or haptic devices.

## ACKNOWLEDGMENTS

This research is promoted by and conducted in collaboration with the Italian National Worker's Compensation Authority (INAIL), under the project "Sistemi Cibernetici Collaborativi - Robot Tele-operativo 2", and supported by Basic Science Research Program through the National Research Foundation of Korea (NRF) funded by the Ministry of Education (NRF-2022R1A6A3A03069040).

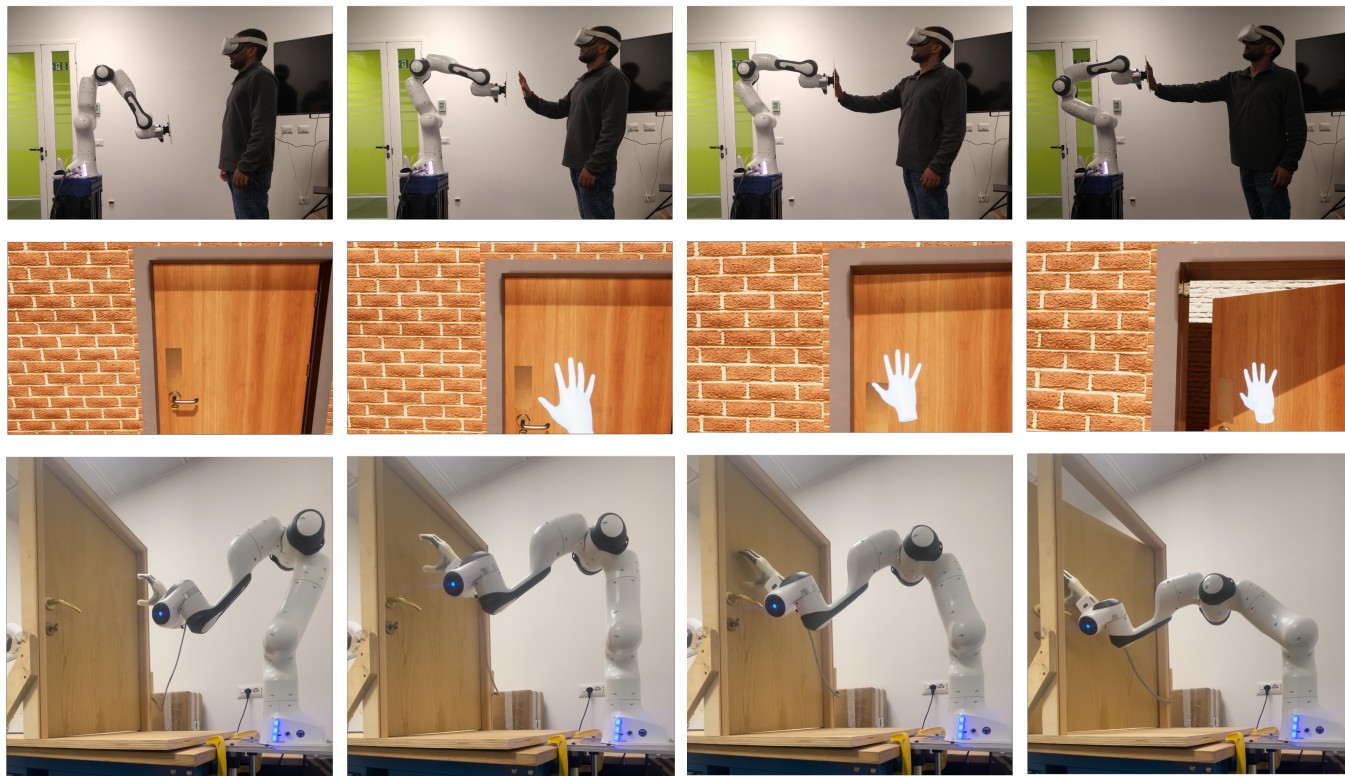

**Figure 3: Interaction with immersive ETH-Telemanipulation setup**

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

Received 18 February 2023; revised 7 March 2023