# OpenReview forum: "Towards Encountered-type Haptic Interaction for Immersive Bilateral Telemanipulation"
_humanrobotinteraction.org/HRI/2023/Workshop/VAM-HRI — VAM-HRI 2023 Oral_

### Official Review · Program_Chairs · 2023-02-25
**Accept**

**Rating:** 9
**Confidence:** 5

**Review:**

Review 1:

This work investigates how immersive remote robot teleoperation can be enhanced through bare-hand interaction with visuo-haptic feedback. The authors achieve this by implementing Encountered-type haptics (ETH), which provides users with real physical feedback and world object manipulation capabilities. Future work includes handling runtime mapping algorithms, and a further investigation on the effects of their system.

Strengths:

System Design: The authors clearly explain their system and the capabilities it provides.

Novelty: The authors use the robot to provide both physical feedback to the user and the user’s force feedback mapped to the robot end effector control. The latter adds an interesting layer, and I am interested to see the continuation of this work.

Weaknesses:

Future work: I would have appreciated to hear more about the drawbacks of this system and to see how the authors plan to tackle the challenges in the future. For example, mapping one hand to manipulation and the other hand for navigation seems limiting for tasks that require two hands. Although, I understand the current setup only affords single arm interaction. Furthermore, how do the authors plan on incorporating objects that don’t provide a flat surface (for example: balls, bottles, door handles)?

Side note:
[1] is a reference to a paper that is extremely relevant to the authors work, and should be cited in the paper.

[1] Ryo Suzuki, Hooman Hedayati, Clement Zheng, James L Bohn, Daniel Szafir, Ellen Yi-Luen Do, Mark D Gross, and Daniel Leithinger. 2020. Roomshift: Room-scale dynamic haptics for vr with furniture-moving swarm robots. In Proceedings of the 2020 CHI Conference on Human Factors in Computing Systems. 1–11. https://doi.org/10.1145/3313831.3376523

The authors will benefit from talking about their current and future work at VAM-HRI, therefore I would argue for it to be accepted.

Review 2:

This paper successfully demonstrates a fully designed system that allows a person to remotely teleoperate a robot via haptic interaction in VR. This system has potential applications across many domains, depending on the fabricated object attached to the manipulator robot and the virtual environment shown to the user.

Formatting – Make sure images are placed logically without leaving single lines of text at the top or bottom of a column (see top of pg 3).

Figure 3 is fantastic – this could even be made larger and placed earlier in the paper, such as when the system is being described in Section 2. This image is a great example of the system in use.

Recommend strong accept.

---

### Decision · Program_Chairs · 2023-03-02

Accept (Oral)